# Opinion: On the Way towards the New Paradigm of Atherosclerosis

**DOI:** 10.3390/ijms23042152

**Published:** 2022-02-15

**Authors:** Alexander A. Mironov, Galina V. Beznoussenko

**Affiliations:** Laboratory of Electron Microscopy, The FIRC Institute of Molecular Oncology, 20139 Milan, Italy; galina.beznusenko@ifom.eu

**Keywords:** atherosclerosis, endothelial cell, enterocyte, Golgi apparatus, chylomicron, modified LDL, glycosylation of apo-protein

## Abstract

Atherosclerosis is a multicausal disease characterized by the formation of cholesterol-containing plaque in the pronounced intima nearest to the heart’s elastic-type arteries that have high levels of blood circulation. Plaques are formed due to arterial pressure-induced damage to the endothelium in areas of turbulent blood flow. It is found in the majority of the Western population, including young people. This denies the monogenic mechanism of atherogenesis. In 1988, Orekhov et al. and Kawai et al. discovered that the presence of atherogenic (modified, including oxidized ones) LDLs is necessary for atherogenesis. On the basis of our discovery, suggesting that the overloading of enterocytes with lipids could lead to the formation of modified LDLs, we proposed a new hypothesis explaining the main factors of atherogenesis. Indeed, when endothelial cells are damaged and then pass through the G2 phase of their cell cycle they secrete proteins into their basement membrane. This leads to thickening of the basement membrane and increases its affinity to LDL especially for modified ones. When the enterocyte transcytosis pathway is overloaded with fat, very large chylomicrons are formed, which have few sialic acids, circulate in the blood for a long time, undergo oxidation, and can induce the production of autoantibodies. It is the sialic acids that shield the short forks of the polysaccharide chains to which autoantibodies are produced. Here, these data are evaluated from the point of view of our new model.

## 1. Introduction

Atherosclerosis is a non-monogenic, diet-related disease characterized by an accumulation of cholesterol in the intima of the human elastic-type arteries and the formation of intimal plaques, namely, bulging intima containing a lot of cholesterol within the foam cells, which originate from macrophages and smooth muscle cells (SMCs). It is necessary to distinguish atherosclerosis from arteriosclerosis [1]. This disease became almost universal because atherosclerosis was found in the majority of the population, including young people. This denies the monogenic mechanism of atherogenesis. At the age of 10–19 years atherosclerotic formations in coronary arteries are found in 65% of males and 62% of females. In the third decade of life, only 11–12% of men and women do not have atherosclerotic changes. Moreover, 46% of men and 33% of women have fibrous plaques, and by 35–45 years atherosclerotic formations are found in the arteries of the brain. In the fourth decade, only 4% of men and 7% of women do not have atherosclerotic changes, and 3–4.5% have calcinosis [2,3,4].

According to the current consensus, atherosclerosis is a multicausal disease involving many physiological and pathological mechanisms. Among the factors currently established are genetic predisposition, hemodynamic conditions in certain parts of the vascular bed, combinations of various risk-factors (hypercholesterolemia, arterial hypertension, diabetes mellitus), immune and autoimmune disorders, viral infection, etc. [5,6,7,8,9,10]. Retention and subsequent accumulation of the low-density lipoproteins (LDLs) in the artery wall triggers a number of events that initiate and propagate lesion development [11,12]. In the human aorta, the percentage of stellate cells in an atherosclerotic lesion considerably exceeds that of the normal intima. In addition, thinning and arborization of contact-forming cellular processes were observed in the fatty streaks [13]. It is thought that the SMCs migrate to the intima from the middle sheath, but in large animals, including humans, their migration here is not required, because SMCs are already present in the human intima. [14,15,16,17,18]. Intimal cells lose contact with each other and, accumulating cholesterol in lysosomes rather than in ER, turn into foam cells filled with lipid droplets formed from lysosomes. Under normal conditions, a common pathway for the formation of lipid granules, especially enlarged ones, is the transformation of lipids from smooth ER into lipid droplets [19]. It is usually difficult to answer the question of what was the main cause of disease development. However, if you remove genetics, the main cause of its development is improper nutrition (see below).

## 2. History

The study of atherosclerosis has a long history. In the early 1850s, M. Rokitansky suggested that fibrin deposition in areas of the damaged vascular wall could damage the vessel. At the end of the 19th century, R. Virchow pointed out that the accumulation of lipids in the vascular intima promotes cell proliferation and development of plaques. In 1904, F. Marchand discovered the accumulation of lipids in atherosclerotic plaques observed in the walls of elastic-type arteries and proposed the name “atherosclerosis” [20].

In 1907, Ignatowski began to feed rabbits a diet of full-fat milk, eggs, and meat. Rabbits fed with animal proteins soon developed pronounced atherosclerosis of the aorta. In 1908 he published his pioneering work on his research [21]. However, the explanation of the mechanisms of this process was incorrect and linked to proteins.

In 1913, Anichkov and Chalatow [22] discovered the major role of cholesterol in the development of atherosclerosis. They fed rabbits cholesterol dissolved in sunflower oil and obtained plaques on the walls of the rabbit’s aorta, similar to the atherosclerotic plaques in humans. Anitschkow hypothesized that atherosclerosis is impossible without externally supplied cholesterol and developed a cholesterol model of atherosclerosis in rabbits. He defined atherosclerosis as a chronic disease characterized by primary lipid deposition in the artery wall, secondary reactive overgrowth of connective tissue, and as a result the formation of sclerotic thickenings or plaques [23,24,25]. This achievement is recognized in the USA as one of the 10 most important discoveries in medicine. In an editorial in the *Annals of Internal Medicine*, W. Dock compared the significance of Anitschkow’s classic work with the significance of Robert Koch’s discovery of the tuberculosis pathogen [26]. In 2004, Steinberg [27] wrote: “If the full significance of his findings had been appreciated at the time, we might have saved more than 30 years in the long struggle to settle the cholesterol controversy and Anitschkow might have won a Nobel Prize”.

In 1938–1939, Harbitz and Müller described the human familial hypercholesterolemia [28].

In 1946, Steiner and Kendall achieved atherosclerosis using the Anitschkow-Chalatow model in dogs [29].

In 1949, Gofman et al. [30] discovered the lipoproteins and their correlation with the risk of coronary heart disease.

In 1951, Russ et al. [31] demonstrated that different classes of lipoproteins have different biological functions.

In 1952, Kinsell et al. [32,33] found that blood cholesterol in normal subjects is increased by saturated fats in the diet.

In 1955, an international epidemiologic survey (the Framingham Heart Study) demonstrated that incidence of coronary heart disease directly correlated to hypercholesterolemia and to dietary fat intake. 

In 1961, it was shown that coronary heart disease risk is highest in groups with the highest blood cholesterol levels [33].

In 1964, K. Bloch received the Nobel Prize for the discovery of the cholesterol biosynthesis pathway [33].

In 1966–1969, it was claimed and then established that reducing blood cholesterol levels by reducing saturated fat in the diet reduced the risk of coronary heart disease [33].

In 1974, the LDL receptor and regulation of cholesterol and lipoprotein metabolism were discovered [34]. The authors (Brown and Goldstein) received the Nobel Prize in 1985 [35,36].

In 1976, Endo et al. [37] discovered the first effective statin.

In 1976, Ross et al. [38] suggested that atherosclerosis is the result of repeated damage of the endothelial cells (ECs) accompanied by platelet adhesion, their activation on the exposed subendothelial surface and macrophage migration into the intima. The result is formation of foamy cells of macrophage origin, migration and proliferation of smooth muscle cells, synthesis and deposition of extracellular matrix, and tissue fibrosis. Nevertheless, the presence of extensive areas of de-endothelization in the endothelial lining was also confirmed [15,39,40,41]. However, the role of endothelial damage is indicated by the absence of atherosclerosis in the pulmonary artery in patients with atherosclerosis, which begins to develop when the pressure in it increases [33].

In 1980, Y. Watanabe [42] produced rabbits with hereditary hyperlipidemia (the WHHL rabbits) [42] expressing the non-functional LDL receptors [43]. He might have been a candidate for the Nobel Prize, however on 13 December 2008, he died at 81 years old.

In 1980, mevinolin (lovastatin), the first commercial statin, was created [33].

In 1981, the role of the autoimmune reaction in the development of atherosclerosis was discovered [44,45].

In 1984, the Lipid Research Clinics’ Coronary Primary Prevention Trial showed a significant reduction in primary coronary heart disease events in men with hypercholesterolemia who received cholestyramine. It was declared that the reduction of blood cholesterol was a national public health goal [33,45].

In 1988, Orekhov et al. [46] (he is the author for correspondence) discovered the role of modified (de-sialylated and with immune complexes) LDLs for the development of atherosclerosis. In the same year, a group of Japanese researchers led by Kawai [47] published evidence that oxidized LDLs also cause atherosclerosis. These two scientists deserved to be the main candidates for the Nobel Prize, even though Prof. Kawai was 91 years old. In the same year it was shown that rabbits in which the natural antioxidants were removed had serious arteriosclerosis (i.e., without the lipid component) [48,49].

In 1992, two groups, namely, led by Breslow [50] and by Maeda [51,52] developed lipoprotein E-deficient mice. Thus, the role of apoproteins (ApoB, ApoE, ApoA), LDL receptors, and the so-called scavenger receptors in the development of atherosclerosis was discovered. These scientists also deserved the Nobel Prize [53].

In 1994, the first large-scale, randomized (based on randomized patient selection), double-blind study (Scandinavian Simvastatin Survival Study) was conducted, which showed that simvastatin treatment not only reduced mortality from coronary heart disease, but also reduced all-cause mortality [33].

In 1995, Steinberg [54] actively developed the idea that oxidized LDL is the main cause of atherosclerosis. However, these ideas were not confirmed in the clinic: high doses of antioxidants did not suppress atherosclerosis whereas the power and the safety of statin intervention was established [33,55,56].

Further, Libby et al. [57] promoted the inflammatory theory of atherosclerosis. They tried to treat atherosclerosis with monoclonal antibodies against a proinflammatory cytokine. 

In 2015, it was shown that reducing the amount of LDL led to a reduction in the severity of atherosclerosis, but attempts to influence atherosclerosis by increasing the amount of HDL had no therapeutic effect [58].

In 2020, a new hypothesis emerged [53,59]. It has been shown that when enterocytes are overloaded with fat, the fat droplets (chylomicrons) produced in the secretory pathway of small intestinal cells increase in size and contain fewer proteins, although they are sucked into the lymph [60,61]. When such chylomicrons pass through the overloaded Golgi complex (a cellular organelle where a long chain of polysaccharides, like starch or glycogen, is attached to proteins for export) glycosylation (attachment of such chains) of apoproteins is disrupted. Glycosylation errors lead to autoantigens, to which the body responds by producing autoantibodies [62].

## 3. Our Scientific Discoveries

Our studies of atherosclerosis were performed similarly to the description presented by Lakatos in his theory of the scientific program [63]. In 1985, we began our study of atherogenesis [64]. At that time, the endothelial damage and the excessive cholesterol uptake from food were considered the main factors of atherogenesis. Initially, we decided to check why LDLs attach more strongly to subendothelial structures in areas of turbulent blood flow, where the endothelium is most often damaged and divided in order to close the defect. For this purpose, the commonly used assay based on de-endothelization of the aorta using a balloon catheter turned out to be unsuitable, and we developed a method of de-endothelization by cryo-damage, which also made it possible to investigate large defects of the endothelial lining, since the phenomenon of covering the luminal surface with smooth myocytes, which was detected after mechanical de-endothelization, was absent. It turned out that the greater ability of the subendothelial layer to capture LDL in places of multiple injuries is due to the fact that after multiple injuries and regeneration of the endothelium, a thick multilayer basement membrane is formed, which more strongly binds LDL [64,65,66,67]. The synthesis and secretion of basement membrane proteins occurs during the G2 phase [68]. In addition, it was found that the thickening of the basement membrane leads to an acceleration of the movement of the endothelium during its regeneration, but after regeneration, the endothelium peels off faster under the mechanical action of turbulence of the blood flow from the places where the basement membrane is thickened [69].

Then we studied why many features of atherosclerotic plaque found in humans cannot be replicated in small animals and proved that this is due to the fact that the intima in humans, like in other large animals, contains pericyte-like cells and even smooth myocytes. Therefore, there is no need for migration of SMCs from the media, which was observed in the arteries of small animals when modeling atherosclerosis [14,69].

Then, in 1988, it was found that modified LDL, including oxidized LDL, play an important role in atherogenesis [46,47,70]. The question arose how these modified LDL are formed. It became necessary to understand why a person suffering from atherosclerosis, even vegans who do not eat animal products and whose body does not receive external cholesterol, have modified LDL. To do this, it was necessary to study the function of the Golgi apparatus, which glycosylates proteins in hepatocytes, which are part of the VLDL, and proteins contained in chylomicrons, which are formed in enterocytes of the small intestine. The fact is that the modified LDL had a low content of sialic acid in polysaccharides synthesized on apo-proteins. We studied the function of the Golgi apparatus [71], and then checked how the lipid overload of enterocytes affects the structure and chemical composition of chylomicrons [60]. Further, it was proved that enterocyte overload can stimulate the synthesis of polysaccharide autoantigens [62].

The three main factors of this hypothesis are: (1) damage to the endothelium and its synthesis of a multilayer basement membrane in places where it is most often damaged, and this basement has the property of binding more strongly to LDL; (2) intima, which, unlike the intima of small animals usually used for modeling atherosclerosis, is multicellular; (3) violation of glycosylation of chylomicrons by enterocytes when the latter are overloaded with lipids; and (4) the excessive consumption of cholesterol and lipids especially when these substances were oxidized. The role of the familiar genetic defects remains important as factors that potentiate atherogenesis [53,59]. Here we would like to analyze these factors point by point, although without the genetic ones. The full hypothesis will be presented in the Conclusion section.

## 4. Role of Intima

Several animal models including rodents (mice, rabbits, rats, hamsters, guinea pigs), avian (pigeons, chickens, quail), swine, carnivora (dogs, cats), and non-human primates were proposed [72]. In order to explain atherogenesis, several hypotheses were formulated [73]. However, all these hypotheses do not consider the role of the specific features of human intima. Indeed, the important feature of human atherosclerosis is the almost complete absence of similar disease in other mammals especially of those with small size, where the intima in most of the arteries is composed of only endothelial cells and their reticular basement membranes [53,64]. On the other hand, atherosclerosis has not been described even in large animals, in which, unlike small animals, intact intima has a structure similar to that in humans and does not consist of only endothelium, reticular basement membrane and separate loci of elastic fibers and collagen fibrils as in small animals, although arteriosclerosis, but not atherosclerosis, was found in giraffes and elephants (reviewed by [53]). In old dogs, one could find arteriosclerosis, but not atherosclerosis [74,75]. In giraffes and elephants, lesions were of two types: intimal arteriosclerosis and medial calcific sclerosis [76,77,78,79]. In elephants, arteriosclerotic plaques are similar to those in human arteriosclerosis [76]. Most of animals do not eat a lot of lipids especially after their prolonged storage in the presence of oxygen. Small animals are not subjected to atherosclerosis. Rats and mice are particularly resistant. Their intima consists of the monolayer of endothelial cells and their reticular basement membrane [53,64,80,81,82,83].

Steiner and Kendall [84,85,86] showed that after irradiation with X-rays, the feeding of dogs with a large amount of cholesterol leads to the development of atherosclerosis. Human muscle arteries are not subjected to atherosclerosis. Atherosclerosis-resistant arteries of the muscular type form minimal to no intimal hyperplasia [87,88,89]. In humans only the arteries of the elastic type were affected but only if the pressure in their lumen is sufficient for endothelial damage (the case of pulmonary artery).

The structure of the intima in large arteries in humans and rabbits is different [14,90,91,92,93]. The intima of the human aorta is filled with pericyte-like stellate cells. Most (84–93%) of the intimal cells exhibit antigens of SMCs and pericytes [14,15,65,66,67,69,94,95,96]. Pericyte-like cells have been identified in the inner intima [97]. The intimal SMCs synthesise collagen I [82]. Cultures of subendothelial cells from the human aortic intima that contained a mixed cell population was made up mainly of typical and modified smooth muscle cells [98].

Conversion of lipids into pericytes or smooth muscle cells is very difficult because these cells are surrounded with basement membrane. Thus, it is necessary to eliminate the basement membrane in order to give their plasma membrane a possibility of uptaking lipids from the interstitial space. One of these possibilities could be the break of intercellular contacts [14,95].

## 5. Role of Endothelial Damage

The hypothesis of atherogenesis proposed by Ross [38,95] poses that atherosclerosis is the result of repeated damage of the endothelium, accompanied by platelet adhesion, their activation on the exposed subendothelial surface, and the migration of macrophages into the intima. This results in the formation of macrophage-derived foam cells, the migration and proliferation of SMCs, the synthesis and deposition of extracellular matrix, and tissue fibrosis. However, the presence of extensive areas not covered with endothelium in the endothelial lining has been refuted [15,39,40,41]. In the human aorta fixed immediately after death, the ECs localized within the sites of turbulent blood flow often contain cilia [15,99,100]. The percentage of endothelial cells with cilia was higher in cells taken from plaques [101] Regeneration of aorta endothelium depends on the damage of the intimal basement membrane and turbulent blood flow. The important role of the endothelium is confirmed by experiments with toxins, radiation damage to the endothelium, pressure, and age. Aging, like hypertension, leads to more frequent endothelial cell damage, cell division, and synthesis of basal membrane proteins and deposition of extracellular matrix proteins under the arterial elastic endothelium. Toxins such as nicotine also potentiate atherogenesis [64,68,69,80,101,102,103,104,105].

Atherogenesis includes the arterial pressure-induced damage to the endothelium in areas of turbulent blood flow, which is exacerbated when it is exposed to toxins. More frequently this happens to the endothelial cells in the G2-phase, when endothelial cells secrete proteins of basement membrane, during chronic hypertension, repeated damage, and aging which leads to a thickening of the basal membrane and increases its affinity for LDL, especially modified. A multilayer basement membrane is formed [53,61,106].

Importantly, during the first 16 days of feeding 2% cholesterol of normolipidemic rabbits, focal increases in arterial LDL concentration precede development of fatty streak lesions [39]. Permeability to LDL did not increase in any aortic site during the 16 days of cholesterol feeding, even in sites with the largest increases in arterial LDL concentrations [40]. This suggests that the level of LDL binding to the subendothelial extracellular matrix is more important than permeability.

As in areas of turbulent blood flow, the ECs are more likely to die and regenerate, this is where more modified LDL attaches to the hypertrophic intimal matrix. Using a model of cryo-damage of the rat aortic wall we have shown that ECs in C phase, which is necessarily followed by G2 phase, are more often found in areas of turbulent blood flow. After repeated de-endothelization the BM becomes multilayered. Similarly, basement membranes becomes multilayered in rats with genetically determined arterial hypertension and with hypertension caused by renal salt overload (Figure 1). Finally, old rats also accumulate BM substance. We found that after cryo-destruction of the medial membrane SMCs (which occurs simultaneously with endothelial destruction), the SMCs from undamaged zones migrate into the subendothelial layer forming spindle-shaped protrusions with their tip directed towards the cryo-damaged zone, which is already covered by regenerated endothelium [107]. In the zone of de-endothelization, which was formed after repeated cryo-damage of the abdominal aortic wall, there was more attachment of large chylomicrons and LDL of blood taken from rats, in which enterocytes were overloaded with fat [16,108,109,110,111] (Figure 1).

## 6. Role of Modified LDLs

It is easier to induce atherosclerosis if oxy-cholesterol is added. In contrast, it is more difficult to obtain atherosclerosis if the cholesterol was additionally purified [112]. Feeding of rabbits with a cholesterol preparation containing 3–5% of cholesterol autooxidation products promotes elevation of plasma cholesterol and atherogenic low- and very-low-density lipoproteins as well as an accumulation of neutral lipids (largely, of cholesterol ether) in hepatocytes and intramural arteries of the myocardium. [112]. The similar dose of non-oxidized cholesterol did not induce marked or any changes at all in rabbits’ lipid metabolism and aortic status [113]. However, antioxidants do not work if high cholesterol levels are maintained [54,114,115,116,117]. The small dense LDL subclass includes an electronegative LDL species associated with endothelial dysfunction [118]. These LDL contained lower levels of sialic acid [46,119,120]. The incubation of cultured human aortic subendothelial cells with de-sialylated LDL, LDL immobilized on latex, and LDL-free latex microspheres induced the alterations in cell-to-cell contacts similar to those occurring in a fatty streak in situ [66,67]. The trans-sialidase, which has enzymatic activity, was isolated from human blood plasma [121,122,123]. However, cDNA of the protein responsible for this function is not yet presented. Decreasing LDL enzymatic de-sialylation could prevent lipid accumulation [123]. LDL-containing circulating immune complexes also play a role in atherogenesis [122]. Importantly, estrogens significantly inhibit the LDL transcytosis by down-regulating endothelial SR-BI and the LDL binding to the endothelial multilayered basement membrane [124]. The atherogenicity of LDL is linked to the ability of its apoB100 moiety to interact with arterial wall proteoglycans [11,125].

## 7. Role of Overloading of Enterocytes with Lipids in the Formation of Modified LDLs

Taking into consideration that atherosclerosis also occurs in vegans, we tested the effect of different doses of introduced fats in the intestinal lumen on the size and glycosylation of chylomicrons. It turned out that when enterocytes are overloaded with fat, the process of lipid transcytosis through enterocytes is disturbed, lipid droplets associated with ER appear, chylomicrons become larger and their level of sialylation, sitting on the binding of the corresponding lectins, is reduced. Moreover, this overload leads to the synthesis of polysaccharide antigens on apo-proteins that are not formed normally and, as a consequence, apparently causes the synthesis of auto-antibodies. It is also known that enterocytes cannot synthesize sufficient amounts of cholesterol themselves, but are able to capture it from blood LDL, which are formed from VLDL, formed by hepatocytes. It is the consumption of too much vegetable fat in one meal that leads to vegans forming large chylomicrons, which circulate longer in the blood and are more susceptible to oxidation in this oxygenated liquid [60,61]. Lipid transcytosis is impaired when enterocytes are overloaded with lipids, chylomicrons are increased in size [60]. They have glycosylation defects leading to the formation of autoantibodies [62]. When enterocytes are overloaded with lipids, the sialylation of apoproteins and lipids is disturbed and the short branches at the ends of polysaccharide chains are formed from monosaccharides, which can cause auto-antibodies to form. It is the sialic acids that shield the short forks of the polysaccharide chains to which the autoantibodies are produced. These enlarged ChMs have few sialic acids at the ends of their polysaccharide chains attached to ApoB [62]. Both small and large lipid particles are actively absorbed from the interstitial space into the lymphatic capillary lumen via the intra-wall valve. These very large chylomicrons enter the lymph, and some of them are retained by lymph node macrophages. Large chylomicrons and their remnants circulate longer in oxygenated blood, presumably because their size blocks their transcytosis through the endothelial cells. Therefore, they can be oxidized [61,126] (Figure 2). The residence time of LDL in the circulation is the critical factor in the relationship between plasma LDL subclass level and atherosclerosis risk [127,128].

Lipoproteins can cross even normal endothelium [12,128,129]. However, the molecular mechanisms controlling this process are still not fully understood [130]. This may be one of the mechanisms of formation of modified LDL, the main factor that causes atherosclerosis. Apparently, the process of glycosylation of chylomicrons is disturbed, as it was found with the glycation of LDL in non-diabetic people: small dense LDL is preferentially glycated both in vivo and in vitro [131]. Fractional feeding of rabbits with cholesterol reduces the growth rate of atherosclerotic plaques (our preliminary unpublished observations). Tight junctions in the lymph capillaries of the intestine villi protest against diet-induced obesity. Genetic impairment of the function of the lymph capillary intramural valve induces weight loss [132].

## 8. The role of Improper Nutrition

Although the role of food is well-known, new data allow the evaluation of these facts from another point of view. Atherosclerosis became ubiquitous in developed countries (Russia is the record-breaker) when the proportion of meat and animal fats in the diet increased dramatically. In Russia, the death curve from heart and vascular diseases went from the usual European level of 599 deaths per 100 thousand people in 1990, to a previously unknown figure of 927 deaths per 100,000 population in a short period of time, in 2002. It was reduced by more than 20% by 2013, from 927 to 729 deaths per 100,000 inhabitants [133].

A change of diet, such as a Chukchi or an Eskimo moving to the city, dramatically increases his chances of getting atherosclerosis. In Russia, a southerner’s departure to the north leads to the same thing. The fact is that people in the North mostly eat storable food, which apparently has a lot of oxidized cholesterol. After the start of statins and food control for cholesterol, the incidence of atherosclerosis began to decline [53].

Nutritional features, especially increased consumption of oxidized cholesterol and less consumption of antioxidants, contribute to the formation of atherosclerotic plaque. Meanwhile, atherosclerosis also occurs in vegetarians and vegans. To explain this phenomenon, we checked how different doses of fats injected into the intestinal lumen affect the size and glycosylation of chylomicrons. It turned out that when enterocytes are overloaded with fats, the process of lipid transcytosis through enterocytes is disrupted, lipid droplets associated with ER appear, ChMs become larger and their level of sialylation, sitting on the binding of the corresponding lectins, decreases. Moreover, such overload leads to the synthesis of polysaccharide antigens on apo-proteins, which are not formed normally and, as a consequence, apparently causes the synthesis of auto-antibodies [62].

It is also known that enterocytes cannot synthesize cholesterol in sufficient quantities themselves, but are able to capture it from LDL blood, which are formed from LDL formed by hepatocytes [53]. It is the consumption of too much vegetable fat at one meal that leads the same vegans to the formation of large chylomicrons, which circulate longer in the blood and are more susceptible to oxidation in this oxygenated liquid. The beneficial effects of plant-based whole foods, such as fresh fruits, vegetables, dried nuts, flax seeds, whole grains, peas, beans, vegan diets, and dietary fibers in LDL-C reduction and cardiovascular health are summarized [134]. Cholesterol in LDL (LDL-C) levels > 160 mg/dL are associated with >1.5-fold greater risk of CHD than levels < 130 mg/dL [135].

In California, a study of 27,530 Adventists divided into three groups over twenty-one years was conducted. The first group ate a mixed diet, the second group were lacto-vegetarians and the third group were strict vegetarians. The coronary heart disease mortality was 14% lower in the first group than in the general population, 57% lower in lactovo-vegetarians and 77% lower in strict vegetarians. Consequently, atherosclerosis in vegetarians is possible, although rare. No increase in the incidence of atherosclerosis was found in the village of Borovoie in the Ivanovo region, where the inhabitants constantly consumed fresh pork in large quantities [66]. Importantly, the statin treatment reduced the five-year incidence of major coronary events and stroke [136].

## 9. Conclusions

Thus, the consequence of events could be the following: the endothelium of elastic arteries, located in areas of turbulent blood flow, is damaged mechanically by the action of arterial pressure or other damaging factors. This induces the reduction of its barrier properties, either by detaching cells from the strata or by opening its tight junctions in the cell contacts. Multiple entries of endothelial cells into the G2 phase of the cell cycle induces secretion of the multilayered basement membrane which is more prone to LDL binding. Our experiments with cryo-damage of the aorta of Watanabe rabbits supported our hypothesis [137,138,139]. On the other hand, when people eat a lot of lipids their enterocytes are subject to overloading. Glycosylation of chylomicron is impaired and even autoantigens could be formed. Chylomicrons are transformed into LDL which circulate in the blood longer and subjected to oxidation. Lower number of sialic acids on the ends of polysaccharides synthesizes on the amino acid chain of apo-proteins and make these LDLs more able to bind the basement membrane below the endothelial cells. LDL penetrates the intima through the endothelium and attaches to the multilayered basement membrane. Modified LDLs have higher affinity to this matrix, either including LDLs with less sialic acid in polysaccharides synthesized on apoproteins as they pass through the cell secretory pathway (this reduces the negative charge of lipid particles), or containing oxidized lipids or trapped autoantibodies. The data presented in the current paper suggest that prevention of atherosclerosis should be based on eating fresh food and avoiding foods that have been in contact with air for a long time, especially those containing cholesterol and avoiding situations in which large quantities of fat, even of vegetable origin, are eaten in one meal (Figure 3). However additional studies are necessary in order to clarify remaining unclear issues. Thus, there is a working hypothesis that can explain most of the phenomena of atherogenesis. However, additional experiments to test it are required, following the recommendations by Popper, who believed that any hypothesis should formulate observations that, if discovered, would reject it. The prohibiting observation for our hypothesis would be the absence of the effect of fractional (as opposed to once a day) feeding of rabbits with cholesterol on the plaque growth rate.

Hepatocyte (1) synthesizes VLDL (2), which moves into the blood through pores in the endothelial cells of the liver sinusoid (red arrows). Next, VLDL arrives at the blood capillary with continuous endothelium and gives cholesterol to the APM of ECs. Unfolded ApoA1 is synthesized by hepatocytes and also arrives at this capillary (yellow arrows). It passes the endothelium through intercellular contacts and appears in the interstitial space. There, it contacts with the BLPM of ECs and takes cholesterol there, forming HDL. ChM (3) is formed by enterocytes (4). It is transported to the interstitial space and then to the lumen of lymph capillary and delivered to the blood capillary lined with the continuous endothelium (dark green arrows). In the capillary, VLDL and ChM contact with the APM (black double-sided arrows) and insert cholesterol and fatty acids into it. After this, ChM and VLDL are transformed into LDL (5, 6), which are delivered to hepatocytes passing through pores in sinusoidal endothelium. Finally, LDLs are taken by LDL receptors on the PM of hepatocytes. Part of LDL contacts with ECs, lining the aorta in the areas of high hemodynamic stress and can pass into the intima there through leaky contacts (7). Apo1 passes the endothelium through intercellular contacts and appears inside the interstitial space. It contacts with the baso-lateral PM of ECs and takes cholesterol and fatty acids from it. Then, ApoA1 is transformed into HDL of two types (8, 9). Next, it interacts with the PM of tissue cell and takes cholesterol when it is necessary. After this, HDLs are absorbed by lymph capillaries and delivered to the blood (yellow arrows). Finally, HDLs pass through pores of sinusoidal endothelium and reach scavenger receptors of the PM of hepatocytes. Abbreviations: APM, apical PM; ChM, chylomicron; HDL, high-density lipoprotein; LDL, low-density lipoproteins; PM, plasma membrane; VLDL, very LDL. Image (3) is adapted from Mironov et al. [53].

## Figures and Tables

**Figure 1 ijms-23-02152-f001:**
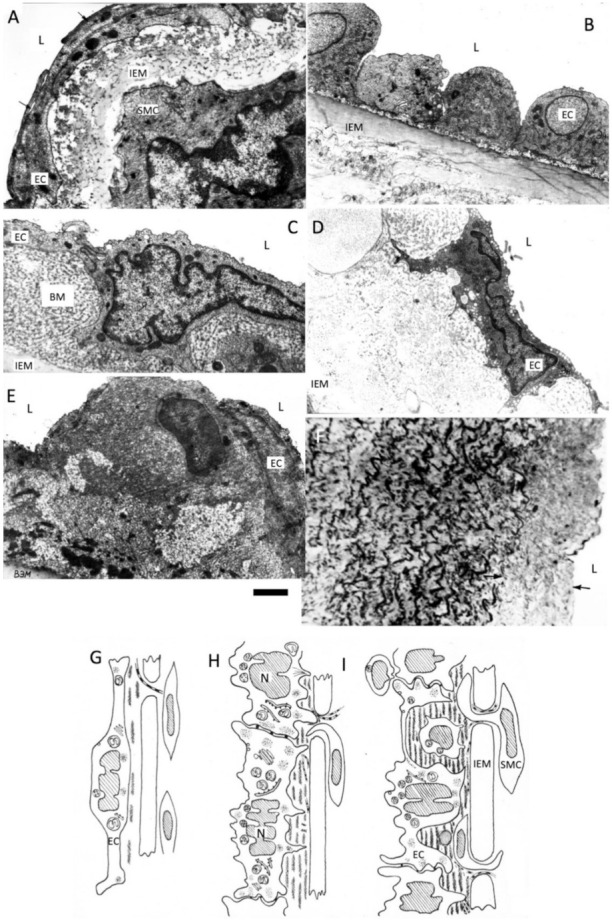
Accumulation of the basement membrane layers after multiple regeneration of endothelium. BM, basement membrane; L, Lumen of aorta; EC, endothelial cell; IEM, internal elastic membrane; SMC, smooth muscle cell. (**A**) Intima of the abdominal aorta in the normal rat; (**B**) Cross section of the abdominal rat aorta at the level of the regenerating endothelium cross sections of the spindle-like endothelial cells (EC) are shown; (**C**) Thick basement membrane (BM) is formed after several round of endothelial regeneration; (**D**) Thick basement membrane in old (24 years) rats; (**E**) Accumulation of extracellular matrix in the hypertensive rats (18 years old); (**F**) Semi-thin section of the intima (arrows) in the hypertensive rats (see **E**); (**G**–**I**) Scheme shows gradual accumulation of the multilayered basement membrane and migration smooth muscle cells (SMC) in the area of turbulent blood flow in rat aorta. Images (**A**–**F**) are from our archives of already published data. Images (**G**–**I**) are adapted from Mironov et al. [53]. Scale bar: 1.5 µm.

**Figure 2 ijms-23-02152-f002:**
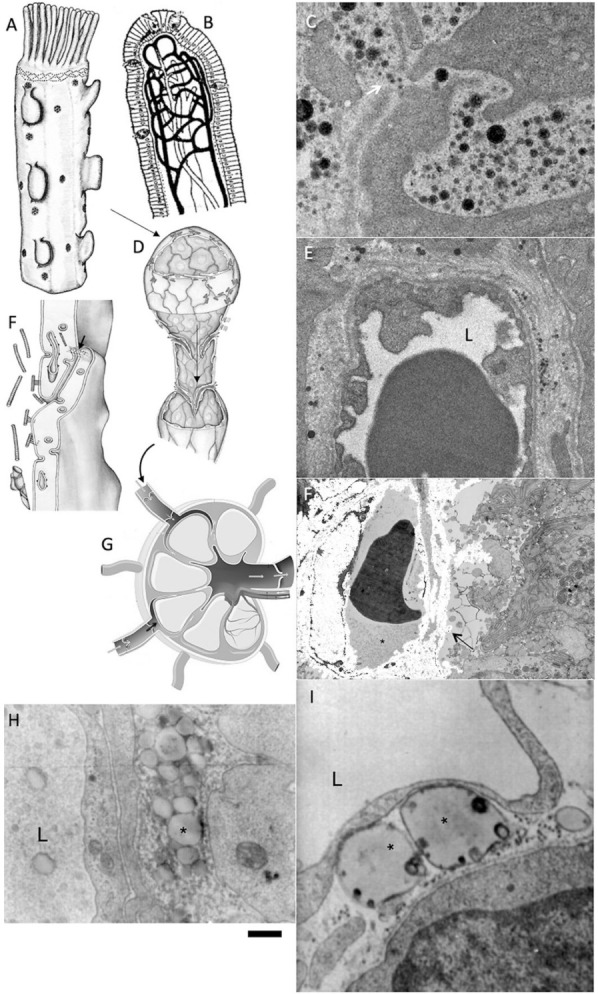
Lipid overloading of enterocytes induces the alteration of chylomicron formation and transport. (**A**,**B**,**D**,**F**,**G**) Schemes of enterocyte (**A**), where chylomicrons are formed, intestine villus (**B**), where chylomicrons are eliminated by lymph capillaries, lymph capillary (**D**), which absorbs chylomicrons, (**F**) intercellular contact in lymph capillary, lymph node (**G**). Chylomicrons pass through the lymph node; (**C**) Passage of normal chylomicrons (black dots) through the whole (white arrow) in the basement membrane; (**E**) Presence of chylomicrons (black and grey dots) in the interstitial space and their absence of the lumen (L) of blood capillary; (**F**) Accumulation of large chylomicrons (arrow) between enterocytes; (**H**) Accumulation of large chylomicrons (asterisk) in the interstitial space and in the lumen (L) of lymph capillary; (**G**) Giant chylomicrons (asterisks) below endothelial cell of the lymph lumen (L). The chylomicron cannot enter the lumen of the lymph capillary. Images (**E**,**F**,**H**,**I**) are from our archives of already published data. Image (**A**) is adapted from Sesorova et al. [60]. Image (**F**) is adapted from Sesorova et al. [61]. Scale bars: 150 nm (**C**); 1 µm (**E**,**F**,**H**,**I**).

**Figure 3 ijms-23-02152-f003:**
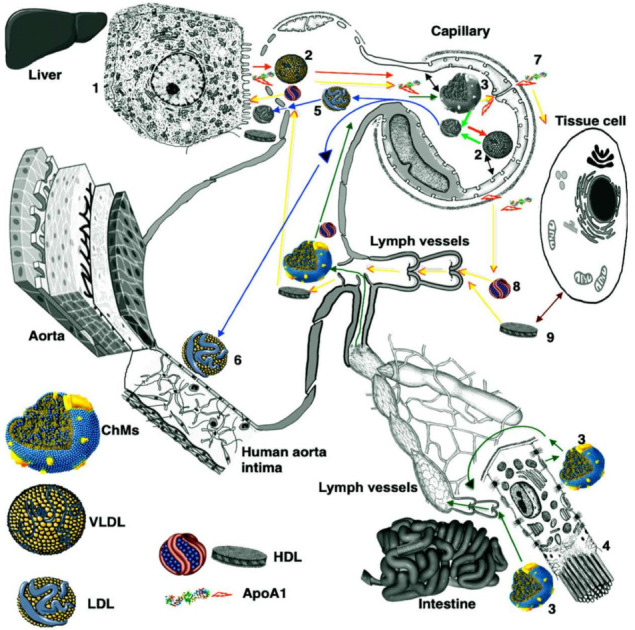
Scheme of LDL and chylomicron circulation and role of enterocytes according to Mironov et al. [53].

## Data Availability

Reported results were taken from already published papers.

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
