# Peer review of "Opinion: On the Way towards the New Paradigm of Atherosclerosis"

_ijms, 2022, doi:10.3390/ijms23042152_

Round 1
Reviewer 1 Report
The review article by Mironov attempts to address the impact (if any) enterocytes have on atherogenesis and atherosclerosis progress. While this concept is novel, appealing, and intriguing, it is in view of the review that the author has missed the mark on the overall rationale of the manuscript. Major and minor comments are highlighted below.
*Major Comments:
*The vast majority of the manuscript is devoted to portions that add little value to the basis of the review. For instance, the "History" section is distracting to the audience and deflects on what the premise of the paper should be. The same for many other manuscript sections as well.
*The part of the paper most linked to the manuscript title is, "Role of enterocytes and their lipid overloading". However, this section is far too subjective with providing self-citations and high praise of this respective work. Since the proposed concepts are relatively new, a more balanced and softened tone compounded with better objectivity is strongly suggested.
*There are numerous statements in other parts of the manuscript that are too subjective and controversial. For instance, claiming certain scientists "deserve" Nobel Prize awards is very distracting and not an appropriate form of scientific communication in a review article, but would be more suitable in a commentary or opinion piece.
*In page 4, it states that, "a new hypothesis has emerged...". I don't think so. I feel that more evidence, along with strong rigor and reproducibility, should be presented for a new hypothesis to be considered before such emphasize can be placed on a potential atherogenic paradigm shift.
*The Conclusion section is too scant and makes bold claims without proper citations to backup certain statements, so this should be addressed.
*Overall, the review article has way too many grammatical errors and incorrect words, statements, and phrases for it to be considered "publication-ready". Therefore, it is highly recommended to have the author request a colleague whose primary language is English to extensively review the manuscript.
*Minor Comments:
*The History section is NOT in chronological order. While the reviewer believes this section should be completely omitted altogether, I still feel that this should be pointed out.
*There are several double spaces throughout the manuscript which need to be corrected.
*It appears that the references are in the wrong citation format, so this should be checked.
*The lone Figure in the manuscript is a nice addition, but the review article could be improved by providing more Figures, particularly ones which illustrate models such as Figure 1. Some material in the manuscript may also be better suited in Table Format instead of providing so much narrative.
Author Response
Major Comments:
1. The vast majority of the manuscript is devoted to portions that add little value to the basis of the review. For instance, the "History" section is distracting to the audience and deflects on what the premise of the paper should be. The same for many other manuscript sections as well.
Reply: We placed some of the history section in the Table. However, did not eliminate it because the second reviewer thinks that this part is very useful for PhD students: “The author has extensive experience in the field and presents very well the history of research in atherosclerosis with the theories on the initiation and development of atheroma plaque. This part of history is very useful not only for the beginners in the field (e.g. master or PhD students).” Therefore, we made Table 1 where the early history is discovered, corrected the order of the history presentation but remained it.
2. The part of the paper most linked to the manuscript title is, "Role of enterocytes and their lipid overloading". However, this section is far too subjective with providing self-citations and high praise of this respective work. Since the proposed concepts are relatively new, a more balanced and softened tone compounded with better objectivity is strongly suggested.
Reply: In the absence of direct and only reliable hypotheses explaining these phenomena and contradictions, scientists can still afford some speculation. The publication of a hypothesis, even if it looks speculative, is very useful for science, since a new working hypothesis appears that could explain the contradictions and which can be tested and either rejected or confirmed. At the same time, in the course of experiments that reject this hypothesis, new more correct hypotheses usually appear. And even if a hypothesis seems speculative to someone, it does not mean that it is wrong, since a dogma in science does not contribute to the scientific progress. And if, where dogma explains all the facts, the speculative nature of the hypothesis allows it to be rejected, then in those areas where there are a dozen hypotheses describing the same phenomenon, or there are no explanations of contradictions at all, even a seemingly speculative hypothesis shifts the search for truth from a dead point. This working hypothesis even if it were incorrect could be tested and even rejected. However, the small rationale of this hypothesis could be useful especially in the areas where many such hypotheses exist. Also, our idea was to make old papers in Russian available for scientific community. There are important observations. In any case, the scientific community could check their conclusions or to take into consideration.
3. There are numerous statements in other parts of the manuscript that are too subjective and controversial. For instance, claiming certain scientists "deserve" Nobel Prize awards is very distracting and not an appropriate form of scientific communication in a review article, but would be more suitable in a commentary or opinion piece.
Reply: We changed the title and shifted our paper to the Opinion type of papers. In order to make our hypothesis look less speculative, we have added a section where we described the history of the creation of our hypothesis of atherosclerosis that is not related to genetic factors. From this section it becomes clear that its creation was carried out within the framework of the theory of the Lakatos’ scientific program.
4. In page 4, it states that, "a new hypothesis has emerged...". I don't think so. I feel that more evidence, along with strong rigor and reproducibility, should be presented for a new hypothesis to be considered before such emphasize can be placed on a potential atherogenic paradigm shift.
Reply: In general, the reviewer is right but his/her objection is valid for theories and lows. However, according to Popper, Kuhn, Laкatos, the hypothesis could be based even on primary experiment or idea. A single observation if this hypothesis is able to explain all facts existing in the field. Our hypothesis can do this. Moreover, we changed the type of the paper and indicated that this is our opinion. In many cases, the so-called speculative hypothesis able to explain contradictions is much more useful that the dogma not able to explain many experimental results. The main task of reviews is to evaluate the probability of incorrect explanations by the authors and to inform scientists about the existence of this so—called speculative hypothesis. According to Lakatos (1978), who analyzed the pathway of the emergence of new paradigms and hypotheses, the initial idea which tries to reject the paradigm could be incorrect. However, during analysis of its contradictions new variants of this model would emerge and these variants could be much more productive than the existing paradigm and the initial rather speculative hypothesis. Strong rigor is not necessary for such hypotheses whereas its reproducibility will be evaluated during consecutive experiments and analysis.
5. The Conclusion section is too scant and makes bold claims without proper citations to backup certain statements, so this should be addressed.
Reply. We re-wrote Conclusions.
6. Overall, the review article has way too many grammatical errors and incorrect words, statements, and phrases for it to be considered "publication-ready". Therefore, it is highly recommended to have the author request a colleague whose primary language is English to extensively review the manuscript.
Reply. We corrected mistakes.
7. Minor Comments: The History section is NOT in chronological order. While the reviewer believes this section should be completely omitted altogether, I still feel that this should be pointed out.
Replay: We changed the order and added Table 1.
8. There are several double spaces throughout the manuscript which need to be corrected.
Reply: We corrected these mistakes.
9. It appears that the references are in the wrong citation format, so this should be checked.
Reply. We corrected mistakes with the help of our Englishman.
10. The lone Figure in the manuscript is a nice addition, but the review article could be improved by providing more Figures, particularly ones which illustrate models such as Figure 1. Some material in the manuscript may also be better suited in Table Format instead of providing so much narrative.
Reply: We added two new Figures and made Table.
Reviewer 2 Report
In the manuscript of the review entitled “Role of enterocytes in atherogenesis”, the author aimed to emphasize the importance of the epithelial cells of the intestine in the atherosclerosis induction/development.
The subject is very interesting, especially since the role of enterocytes in inducing atherosclerosis is less investigated compared to cells of the arterial wall, inflammatory cells or hepatocytes.
The author has extensive experience in the field and presents very well the history of research in atherosclerosis with the theories on the initiation and development of atheroma plaque. This part of history is very useful not only for the beginners in the field (e.g. master or PhD students).
My main concern is the imbalance regarding the content and organization of the manuscript. If the author intends to present an “overview” on atherosclerosis (as he chose to dedicate an important part of the manuscript to “intima”, “endothelium”, “modified LDL”, or “food” contribution to atherosclerosis), then the title of the manuscript does not reflect the content of the script. The contribution of enterocytes to atherosclerosis represents only a small part of the manuscript and addresses mainly processes that take place in lipid overloaded enterocyte, especially the disturbances in glycosylation (sialylation) of apoliproteins and lipids. Or, the enterocytes are involved in many more processes that are part of lipid metabolism (lipid uptake from the intestinal lumen, lipoproteins packing – chylomicrons and HDL- and their secretion, trans-intestinal cholesterol efflux – TICE or bile acid circuit), and are disturbed in pro-atherogenic conditions. The manuscript message is the author’s option, but in the actual form of the script, the message is not very clear.
Minor points to be addressed:
- The manuscript should be checked for typos, see line 147 (double “whereas”), lines 172-174 (repetition of the sentence), lines 218-219 (repetition of “formed from LDL”), line 347 (“of BM” instead of “if BM”), line 392 (“protect” instead of “protest”) and line 413 (small “T”).
- At first appearance in the text the abbreviations should be explained (line 22 – SMC, line 41 – LDL, line 145 – HDL and line 381 – GC).
- The Figure 1 and its legend should be carefully checked for inaccuracies: (i) no. 6 on picture seems to be LDL, not HDL; (ii) the numbers 7, 8 and 9 on picture are not explained in the legend.
Author Response
In the manuscript of the review entitled “Role of enterocytes in atherogenesis”, the author aimed to emphasize the importance of the epithelial cells of the intestine in the atherosclerosis induction/development.
The subject is very interesting, especially since the role of enterocytes in inducing atherosclerosis is less investigated compared to cells of the arterial wall, inflammatory cells or hepatocytes. The author has extensive experience in the field and presents very well the history of research in atherosclerosis with the theories on the initiation and development of atheroma plaque. This part of history is very useful not only for the beginners in the field (e.g. master or PhD students).
My main concern is the imbalance regarding the content and organization of the manuscript. If the author intends to present an “overview” on atherosclerosis (as he chose to dedicate an important part of the manuscript to “intima”, “endothelium”, “modified LDL”, or “food” contribution to atherosclerosis), then the title of the manuscript does not reflect the content of the script. The contribution of enterocytes to atherosclerosis represents only a small part of the manuscript and addresses mainly processes that take place in lipid overloaded enterocyte, especially the disturbances in glycosylation (sialylation) of apoliproteins and lipids. Or, the enterocytes are involved in many more processes that are part of lipid metabolism (lipid uptake from the intestinal lumen, lipoproteins packing – chylomicrons and HDL- and their secretion, trans-intestinal cholesterol efflux – TICE or bile acid circuit), and are disturbed in pro-atherogenic conditions. The manuscript message is the author’s option, but in the actual form of the script, the message is not very clear.
Reply: We shifted our manuscript to the opinion type and changed the title. Our hypothesis was made clearer. We included the information about many additional roles of enterocytes in the manuscript.
Minor points to be addressed:
1. The manuscript should be checked for typos, see line 147 (double “whereas”), lines 172-174 (repetition of the sentence), lines 218-219 (repetition of “formed from LDL”), line 347 (“of BM” instead of “if BM”), line 392 (“protect” instead of “protest”) and line 413 (small “T”).
2. At first appearance in the text the abbreviations should be explained (line 22 – SMC, line 41 – LDL, line 145 – HDL and line 381 – GC).
3. The Figure 1 and its legend should be carefully checked for inaccuracies: (i) no. 6 on picture seems to be LDL, not HDL; (ii) the numbers 7, 8 and 9 on picture are not explained in the legend.
Reply: We corrected these mistakes and decreased the number of abbreviations and added explanations for numbers 7–9 in the Figure 1. Additionally, we introduced two new Figures.
Round 2
Reviewer 1 Report
The manuscript has been extensively changed and modified. However, not all of my initial comments and suggestions stated in my first found of review were incorporated into the manuscript. Once this is accomplished, it might be more publication-ready.
Author Response
“The manuscript has been extensively changed and modified. However, not all of my initial comments and suggestions stated in my first found of review were incorporated into the manuscript. Once this is accomplished, it might be more publication-ready.”
Our reply. We already all changed which the reviewer suggested. Briefly. W changed the type of the paper and now it is an opinion where it is possible to suggest the names of scientists suitable for of future Nobel prize and formulate working hypothesis. The science is not a dictatorship when scientists should fulfill all comments of any reviewer.
Reviewer 2 Report
In the manuscript entitled “Opinion: On the way towards the new atherosclerosis paradigm”, the authors made some major modifications compared to the previous manuscript entitled "Role of enterocytes in atherogenesis".
I still have the following comments for this manuscript:
1. The authors announce a “new paradigm” in the title of the manuscript, but this is not pointed out as such in the abstract. Conclusion presents the chain of events (also exposed in the text of the manuscript) without pointing out “the novelty of the paradigm”.
2. The title “3. Our scientific program” would sound better as “3. Our scientific discoveries” or something like this. The authors mention “the three main factors of this hypothesis are” at the end of this chapter. What the hypothesis is about? The third factor is not highlighted.
3. The manuscript should be checked for typos, see end of chapter 3 and many typos in the figures’ (1 and 2) legends.
Author Response
I still have the following comments for this manuscript:
1. The authors announce a “new paradigm” in the title of the manuscript, but this is not pointed out as such in the abstract. Conclusion presents the chain of events (also exposed in the text of the manuscript) without pointing out “the novelty of the paradigm”.
We indicated the novelty.
2. The title “3. Our scientific program” would sound better as “3. Our scientific discoveries” or something like this. The authors mention “the three main factors of this hypothesis are” at the end of this chapter. What the hypothesis is about? The third factor is not highlighted.
We corrected this.
3. The manuscript should be checked for typos, see end of chapter 3 and many typos in the figures’ (1 and 2) legends.
We corrected these mistakes.